**Investigation**

# Origin and maintenance of large ribosomal RNA gene repeat size in mammals

Emma Macdonald,[1] Annabel Whibley,[1,2] Paul D. Waters,[3] Hardip Patel,[4] Richard J. Edwards,[3,5] Austen R. D. Ganley [1,6,*]

[1]School of Biological Sciences, University of Auckland, Private Bag 92019, Auckland 1142, New Zealand
[2]Grapevine Improvement, Bragato Research Institute, RFH Building, Engineering Drive, Lincoln University, Lincoln 7647, New Zealand
[3]School of Biotechnology and Biomolecular Sciences, UNSW Sydney, Chancellery Walk, Kensington, NSW 2033, Australia
[4]John Curtin School of Medical Research, Australian National University, 131 Garran Rd, Acton, ACT 2601, Australia
[5]Minderoo OceanOmics Centre at UWA, UWA Oceans Institute, University of Western Australia, Crawley WA 6009, Australia
[6]Digital Life Institute, University of Auckland, Private Bag 92019, Auckland 1142, New Zealand

*Corresponding author: School of Biological Sciences, University of Auckland, Private Bag 92019, Auckland 1142, New Zealand. Email: a.ganley@auckland.ac.nz

## Abstract

The genes encoding ribosomal RNA are highly conserved across life and in almost all eukaryotes are present in large tandem repeat arrays called the rDNA. rDNA repeat unit size is conserved across most eukaryotes but has expanded dramatically in mammals, principally through the expansion of the intergenic spacer region that separates adjacent rRNA coding regions. Here, we used long-read sequence data from representatives of the major amniote lineages to determine where in amniote evolution rDNA unit size increased. We find that amniote rDNA unit sizes fall into two narrow size classes: "normal" (~11–20 kb) in all amniotes except monotreme, marsupial, and eutherian mammals, which have "large" (~35–45 kb) sizes. We confirm that increases in intergenic spacer length explain much of this mammalian size increase. However, in stark contrast to the uniformity of mammalian rDNA unit size, mammalian intergenic spacers differ greatly in sequence. These results suggest a large increase in intergenic spacer size occurred in a mammalian ancestor and has been maintained despite substantial sequence changes over the course of mammalian evolution. This points to a previously unrecognized constraint on the length of the intergenic spacer, a region that was thought to be largely neutral. We finish by speculating on possible causes of this constraint.

Keywords: ribosomal DNA; amniotes; mammals; intergenic spacer; IGS; rDNA; ribosomal RNA gene repeats

## Introduction

Ribosomal RNA (rRNA) forms the structural and catalytic core of ribosomes. In most eukaryotes the genes encoding the major rRNAs are organized into a series of head-to-tail tandem repeats that are known as the ribosomal DNA (rDNA; Long and Dawid 1980). This distinctive organization appears to have persisted for over a billion years of eukaryote evolution (Appels and Honeycutt 1986). Transcription of rDNA is also distinctive in eukaryotes as it involves the dedicated RNA polymerase I (Goodfellow and Zomerdijk 2013) and occurs in a specific subdomain of the nucleus, the nucleolus (Sirri et al. 2008).

An inevitable consequence of the tandem repeat organization of rRNA genes is their partitioning into readily definable repeat units. A single rDNA repeat unit consists of a coding region and an intergenic spacer (IGS). The coding region encodes the 18S, 5.8S, and 28S rRNAs (the names reflect sedimentation rates, thus differ slightly between lineages) that are processed from a single transcript (Srivastava and Schlessinger 1991) via the removal of two internal transcribed spacers (ITS1 and ITS2; Nazar 2004). The IGS separates adjacent coding regions and is thought to largely consist of non-functional DNA, which explains the rapid changes in IGS sequence over evolutionary time

(Rogers and Bendich 1987b; Ganley et al. 2005). Nevertheless, some functional elements, including a replication origin (Kwan et al. 2023), a replication fork barrier site (Sasaki and Kobayashi 2022; Hori et al. 2023), and noncoding transcripts (Tsang and Carr 2008; Agrawal and Ganley 2018; Feng and Manley 2022) have been found in a number of characterized IGSs. The IGS also frequently contains sub-repeats: short (~10–200 bp) tandem repeats that can vary in copy number to produce length polymorphisms (Rogers and Bendich 1987b). Sequence variation in the IGS between species is contrasted by high levels of sequence identity between rDNA repeats within a genome (Ganley and Kobayashi 2007). This pattern of high intra-genomic homogeneity is known as concerted evolution, and results from continual turnover of rDNA repeat units, likely by DSB-induced homologous recombination within rDNA (Elder and Turner 1995; Eickbush and Eickbush 2007).

One striking feature of eukaryote rDNA is the narrow range that repeat unit lengths fall into. Most eukaryotes characterized to date have rDNA repeats within a ~2.5-fold size range (~8–20 kb; e.g. Cortadas and Pavon 1982; Karvonen et al. 1993; Mateos and Markow 2005; Voronov et al. 2008; Torres-Machorro et al. 2010). While it is unsurprising that the coding region is relatively invariant in size, the low variation in IGS size is unexpected for a

noncoding region, with IGS length variation being much smaller than the variation in genome length (Pagel and Johnstone 1992). Nevertheless, IGS sizes outside this range are observed, for example, the IGS in the Trichomonad *Tritrichomonas foetus* is only ~200 bp long (Chakrabarti *et al.* 1992), while Eutherian mammals have the longest IGS sizes characterized to date, at ~25–30 kb (Perry *et al.* 1970; Blin *et al.* 1976; Arnheim and Southern 1977; Cory and Adams 1977; Stumph *et al.* 1979).

The large IGS size of mammals compared to other eukaryote lineages has been known for many decades (Perry *et al.* 1970; Blin *et al.* 1976; Arnheim and Southern 1977; Cory and Adams 1977), but the only proposed explanation for this observation we are aware of is a functional relationship to nucleolar substructure. This explanation derives from observations that mammals and all characterized *Sauropsida* (reptile) groups except *Testudines* (turtles) have nucleoli organized into three zones, a phenomenon known as "tripartite" nucleolar organization (nucleolar sub-structure in *Rhynchocephalia* (tuatara) is seemingly not yet described). Conversely, the nucleoli of other lineages are organized into two zones (bipartite organization; Thiry *et al.* 2011). It was suggested there is some functional link between the evolution of tripartite nucleolar organization and increased rDNA unit size (Thiry and Lafontaine 2005). However, as is the case for most eukaryotes, rDNA unit size is poorly characterized in the amniotes (the vertebrate group to which the mammalian and reptilian lineages belong) (Modesto and Anderson 2004), so the evolutionary origin of large rDNA unit sizes remains unclear (see Supplementary Table 1 for examples characterized to date).

Here, we measured rDNA unit size in the major amniote lineages using publicly available and unpublished long-read sequence data to determine when large IGS sizes evolved and whether they are associated with tripartite nucleolar organization. Our results show that rDNA unit size increased at the base of the mammalian clade (i.e. the clade including monotremes, marsupials, and eutherian mammals), with large rDNA unit sizes not found elsewhere in the amniotes. The results also show that widespread changes in IGS sequences have occurred over the course of mammalian evolution despite their relative similarity in size. These results are not consistent with the increase in IGS size being a cause or direct consequence of tripartite nucleolar organization, but they unexpectedly suggest that large rDNA unit size has been actively maintained in mammals.

## Materials and methods

### Datasets

The datasets used in this study are shown in Table 1. SRA files were converted to FASTA format using the fastq-dump command from sratoolkit (v. 2.9.6; Leinonen *et al.* 2011). FASTQ datasets were converted to FASTA format using seqtk (v. 1.3). BAM files were converted into FASTA files using SAMtools (v. 1.10-GCC-9.2.0; Li *et al.* 2009). Low-complexity sequences were masked using a dust-masker (Morgulis *et al.* 2006) as follows:

*<dustmasker -in dataset.fasta -infmt fasta -parse_seqids \ -outfmt maskinfo_asn1_bin -out dataset _dust.asnb>*

### BLAST searching and rDNA unit identification

Nucleotide BLAST (BLAST) searches were performed using the command line BLAST (v. 2.10.0-GCC-9.2.0). BLAST databases were created from masked FASTA files as follows:

*<makeblastdb -in dataset.fasta -dbtype nucl -parse_seqids \ -mask_data dataset.fasta_dust.asnb -out dataset_db -title \ "dataset_database">*

BLAST searches were then performed with the query rDNA sequences listed in Supplementary Table 6 as follows:

*<blastn -db dataset_db -query Query_sequence.fasta -outfmt 7 -out dataset_db_blast.txts>*

ONT reads containing more than one rDNA unit were identified by manual inspection of the BLAST outputs. Positions of rDNA features were identified using the query rDNA unit and sizes were calculated from this as shown in Fig. 1. Histograms and density plots were generated in R-studio using ggplot2 (Wickham 2009) as follows:

*< ggplot(dataset_rDNA_lengths, aes(`rDNA Unit Length (bases)`)) + geom_histogram(aes(y = ..density..), bandwidth = 100, colour="black", fill="white") + geom_density(alpha = .2, fill="#FF6666") >*

Mean and peak rDNA unit sizes were determined as follows:

*<dataset_rDNA_lengths_Mean < - mean(dataset_rDNA_lengths $`rDNA Unit Length (bases)`)>*

and

*<which.max(density((dataset_rDNA_lengths$`rDNA Unit Length (bases)`)$y)>*

*<dataset_rDNA_peak < - density(dataset_rDNA_lengths$`rDNA Unit Length (bases)`)$x[y]>*

For datasets with two peaks, the second peak size was calculated similarly, but with an upper ("< x") or lower x-axis ("> x") limit to exclude the primary peak as follows:

*< Second_MaxY_dataset < - max(density(dataset_rDNA_lengths $`rDNA Unit Length (bases)`)$y[density(dataset_rDNA_lengths$`rDNA Unit Length (bases)`)$x "Primary_peak_x'])>*

### Consensus sequence generation

To generate consensus sequences, 15 rDNA units from the ONT reads were used to determine rDNA unit size, or all rDNA units from the assembly contigs were used. Units were manually edited to start at the 18S rRNA gene and aligned using MAFFT (v. 7.450; Katoh and Standley 2013) in Geneious (v. 2020.05; https://www.geneious.com) with default settings. The resulting consensus sequences were manually annotated following alignment with the corresponding rDNA reference sequence (Supplementary Table 6).

### PacBio rDNA assemblies

Two approaches were used for identifying rDNA units from PacBio data. For the PRJNA513296 dataset (Table 1), files containing PacBio reads were merged using the SAMtools (v. 1.10-GCC-9.2.0) and BLAST was run using the query rDNA unit in Supplementary Table 6. The BLAST output was limited to the 500 lowest *e*-value reads as follows:

*<blastn -db dataset_db -query Query_sequence.fasta -outfmt "6 sseqid" -out dataset_rDNA_readnames.txt>*

BLAST outputs were then sorted to generate a list of the unique read names and those reads were extracted using the seqtk "subseq" command. Flye (v. 2.7.1; Kolmogorov *et al.* 2019) was used to assembly the resultant PacBio reads as follows:

*<flye –pacbio-raw dataset_pacbio_reads.fasta –out-dir Flye_dataset_pacbio_reads_assembly –genome-size Xm –threads 4>*

For the PRJNA433451 dataset, rDNA-containing PacBio reads were identified as follows:

*<minimap2 -H -f 0.00000000000001 –secondary = no -Y -a -x map-pb/g/data/xl04/hrp561/rdnalib/deuterostomia.18s28 s.rDNA.fasta ${i}|samtools view -h -F 4 -b -o/g/data/xl04/hrp561/platy-pus/rdna/aln/$(basename ${i} .fasta.gz).bam>*

Reads with alignment scores >500 were extracted and assembled using Flye (v2.9) as above.

**Table 1.** Datasets used in this study.

| Species | Sequencing method | Database accessed from/dataset contributor | Accession number |
|---|---|---|---|
| *Saccharomyces cerevisiae* (Bakers' yeast) | ONT PacBio | NCBI NCBI | PRJEB19900 PRJEB7245 |
| *Homo Sapiens* (Human) | ONT | Genome in a Bottle Consortium; NCBI | PRJNA200694 |
| *Rhinella marina* (Cane toad) | ONT | Dr Richard Edwards University of New South Wales | N/A |
| *Pseudonaja textilis* (Brown snake) | ONT | Dr Richard Edwards University of New South Wales | N/A |
| *Notechis scutatus* (Tiger snake) | ONT | Dr Richard Edwards University of New South Wales, Australia | N/A |
| *Notiomystis cincta* (Stitchbird hihi) | ONT | Aotearoa Genomic Data Repository | TAONGA-AGDR00034 (https://doi.org/10.57748/ZD00-D451)[a] |
| *Melanerpes aurifrons* (Golden-fronted woodpecker) | ONT | NCBI | PRJNA598863 |
| *Sarcophilus harrisii* (Tasmanian devil) | ONT | NCBI | PRJEB34649 |
| *Sphenodon punctatus* (Tuatara) | ONT | NCBI | PRJNA445603 |
| *Malaclemys terrapin* (Diamondback terrapin) | PacBio | NCBI | PRJNA339452 |
| *Ornithorhynchus anatinus* (Platypus) | PacBio | NCBI | PRJNA513296 and PRJNA433451 |
| *Pan troglodytes* (Chimpanzee) | PacBio | NCBI | PRJEB18078 |

[a]Access is controlled as this is a species of special interest to the Indigenous people of Aotearoa New Zealand (see https://data.agdr.org.nz/ for details and the application process for access requests).

For both assemblies, identification of rDNA-containing contigs was performed using BLAST, and rDNA reference sequence alignment was performed using MAFFT, both as described above.

## Sub-repeat identification

Sub-repeats were initially identified from dotplots of whole rDNA units made within Geneious (v. 2020.05). Manual inspection of sequences was then used to annotate individual sub-repeats and to distinguish microsatellites (unit size <10 bp), which we do not consider here as sub-repeats. Some species showed evidence for sub-repeats from the dotplots, but the sub-repeat units were too dissimilar to reliably distinguish. These degenerate sub-repeats were not annotated or analyzed further.

## IGS matches the genome

The Tasmanian devil and platypus IGSs, after manual removal of sub-repeats, were used as queries to search their respective RefSeq Genome Database genomes using BLASTn. Matches were then used as queries to search the NR databases using BLASTn and BLASTx. In all cases default parameters were used.

## TE identification

Consensus rDNA unit sequences from this study and IGS sequences from human (accession GL00022) and mouse (accession BK000964) were input into RepeatMasker (v. open-4.0.9; http://www.repeatmasker.org/cgi- bin/WEBRepeatMasker) using human, mouse, mammal, or vertebrate as the source reference as appropriate with default parameters. TEs longer than 50 bp were identified from RepeatMasker outputs and nested TEs were reconstructed manually. Potentially orthologous TEs were manually identified via shared TE type and orientation in the way that gave the maximum number.

## Results

We set out to determine where in amniote evolution the large increase in rDNA IGS size seen in eutherian mammals occurred, and whether this correlates with tripartite nucleolar organization. Because the rDNA typically assembles poorly using short-read whole genome sequencing (presumably due to a combination of high copy number and regions that are recalcitrant to short-read sequencing (Agrawal and Ganley 2016; Fan *et al.* 2022)), we focused on long-read sequencing technologies, principally Oxford Nanopore Technologies (ONT) sequencing. Long-read sequencing approaches can overcome these limitations of short-read sequencing because they generate reads that can traverse most or all the rDNA unit length. We started with a simple methodology to determine rDNA unit sizes: identify reads from publicly available ONT datasets that contain more than one rDNA unit using BLAST, then use these multi-unit reads to determine the lengths of the rDNA unit and the elements within (Fig. 1). To verify that this approach could accurately determine rDNA unit sizes, we tested it with species known to have "normal" (~9 kb; *Saccharomyces cerevisiae*) and large (~44 kb, *Homo sapiens*) rDNA units. As expected, rDNA unit size was accurately determined for both species (Supplementary Fig. 1). We also compared the rDNA unit sequence we obtained to that from the recent telomere-to-telomere CHM13 human genome assembly, as this included ONT reads (Nurk *et al.* 2022). The rDNA unit sequences were 85% identical, with the most substantive differences being in Long Repeat regions, and the units are structurally the same (Supplementary Fig. 1). The differences may be a consequence of real sequence variation that the CHM13 assembly was able to resolve by separately assembling rDNA from different chromosomes using a local assembly approach and/or may reflect limitations of our low-coverage approach. Together, these results suggest that our ONT read strategy is capable of accurately determining rDNA unit sizes, but that dedicated, high-coverage approaches such as those

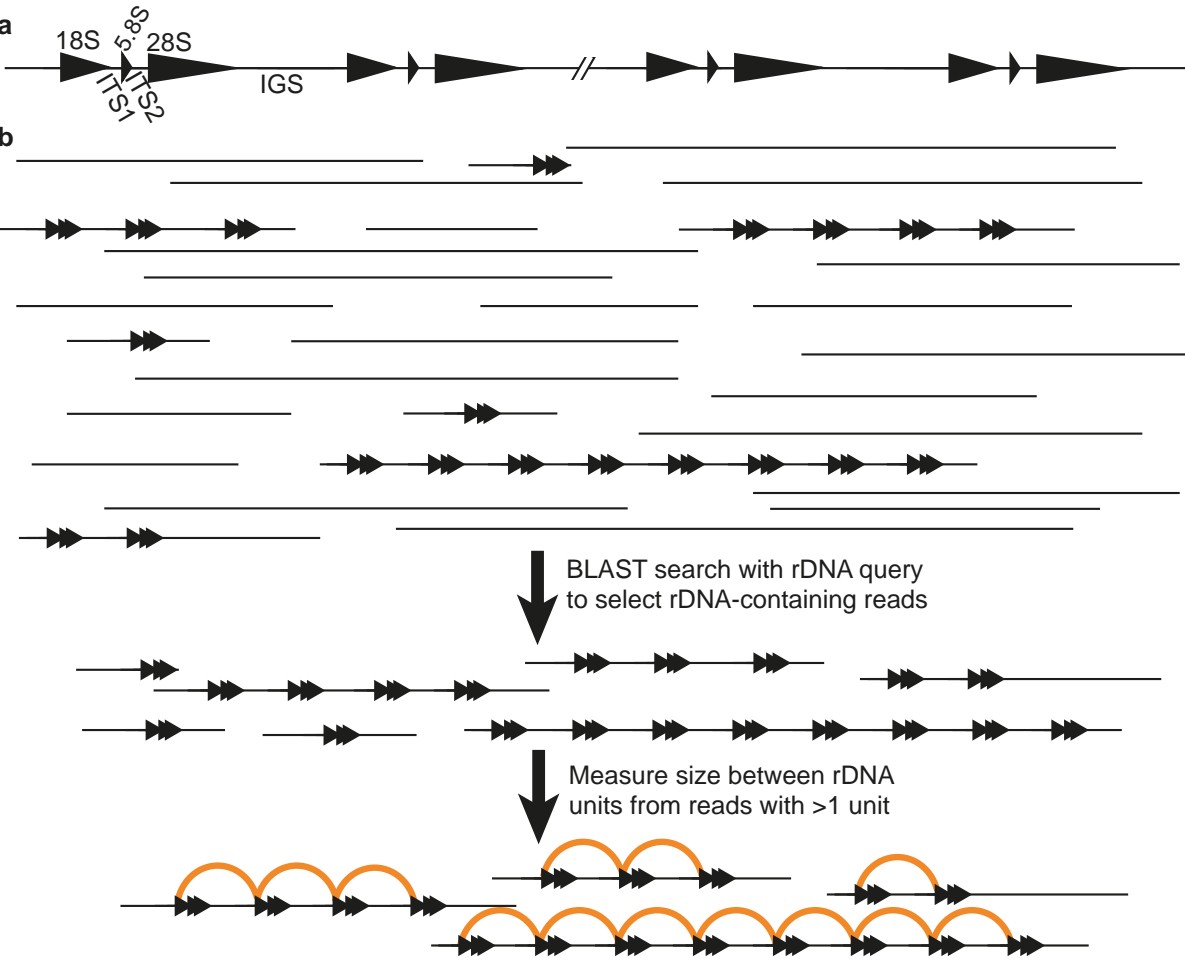

**Fig. 1.** Strategy for determining rDNA unit size from Oxford Nanopore (ONT) sequence data. (a) Schematic showing the organization of eukaryote ribosomal RNA gene repeats. rRNA gene names are shown above; spacer names below. (b) Whole genome ONT datasets (indicated schematically at top) were searched for rDNA-containing reads (rDNA indicated by sets of three arrowheads) by BLAST. rDNA unit sizes were determined from reads with more than one unit by calculating the distance between consecutive instances (half-circles) of the same rDNA feature (e.g. 18S rRNA gene).

employed by the T2T consortium (Nurk *et al.* 2022) are likely needed for the accurate assembly of rDNA sequences.

We used publicly available ONT whole genome datasets from representatives of all major amniote lineages to determine when large rDNA unit sizes originated. Amniotes are split into two groups: *Sauropsida* (reptiles) and *Synapsida* (mammals). ONT datasets were available for representatives of all major clades within the reptiles other than *Testudines* and *Crocodylia*. Extant mammals consist of three clades: *Eutheria* (placental mammals), *Metatheria* (marsupials), and *Monotremata* (monotremes). rDNA unit sizes have previously been reported for several eutherian mammals (Supplementary Table 1) and an ONT dataset was available for marsupials. Finally, an ONT dataset from the cane toad (*Rhinella marina*) in the *Amphibia* (sister group to the amniotes) was used as an outgroup (see Table 1 for details). We applied our BLAST-based method for inferring rDNA unit size to each ONT dataset and characterized rDNA units as being either "normal" (8–20 kb) or "large" (>30 kb) in size.

All reptile species we analyzed, as well as the cane toad, had normal rDNA unit sizes (<20 kb). The sizes we estimate are comparable to those made previously from other reptilian species, suggesting these older size estimates made using restriction-hybridization methods (Supplementary Table 1) are comparable

to those made from sequence data, as previously indicated (e.g. Gonzalez and Sylvester 1995). In contrast, the marsupial (Tasmanian devil; *Sarcophilus harrisii*) had a large rDNA unit size (Fig. 2a; Supplementary Fig. 2; Table 2), similar to characterized eutherian mammal rDNA unit sizes (Supplementary Table 1). We observed variation in rDNA unit sizes between reads, which may result from technical variation due to errors during ONT sequencing (Mikheenko *et al.* 2022) and/or real intra-genomic repeat size variation (e.g. Coen *et al.* 1982; De Lucchini *et al.* 1988). Nevertheless, the standard deviation was less than 10% of the unit size for all species except the Tasmanian devil (Table 2), and this variation does not confound the categorization into normal and large-size classes. In the case of the Tasmanian devil, rDNA reads fell into two different unit sizes: ~40 kb and ~44 kb (Supplementary Fig. 3). The shorter unit size, while containing clearly identifiable 18S and 28S rRNA genes, lacks any identifiable 5.8S rRNA gene, therefore, we used the longer unit for subsequent analyses. However, irrespective of whether the shorter unit represents a variant rDNA unit type or is a sequencing artifact, the rDNA unit size in the Tasmanian devil is large.

The absence of a monotreme ONT dataset limited our ability to determine when in mammalian evolution rDNA size increased. Therefore, we made use of publicly available PacBio datasets for

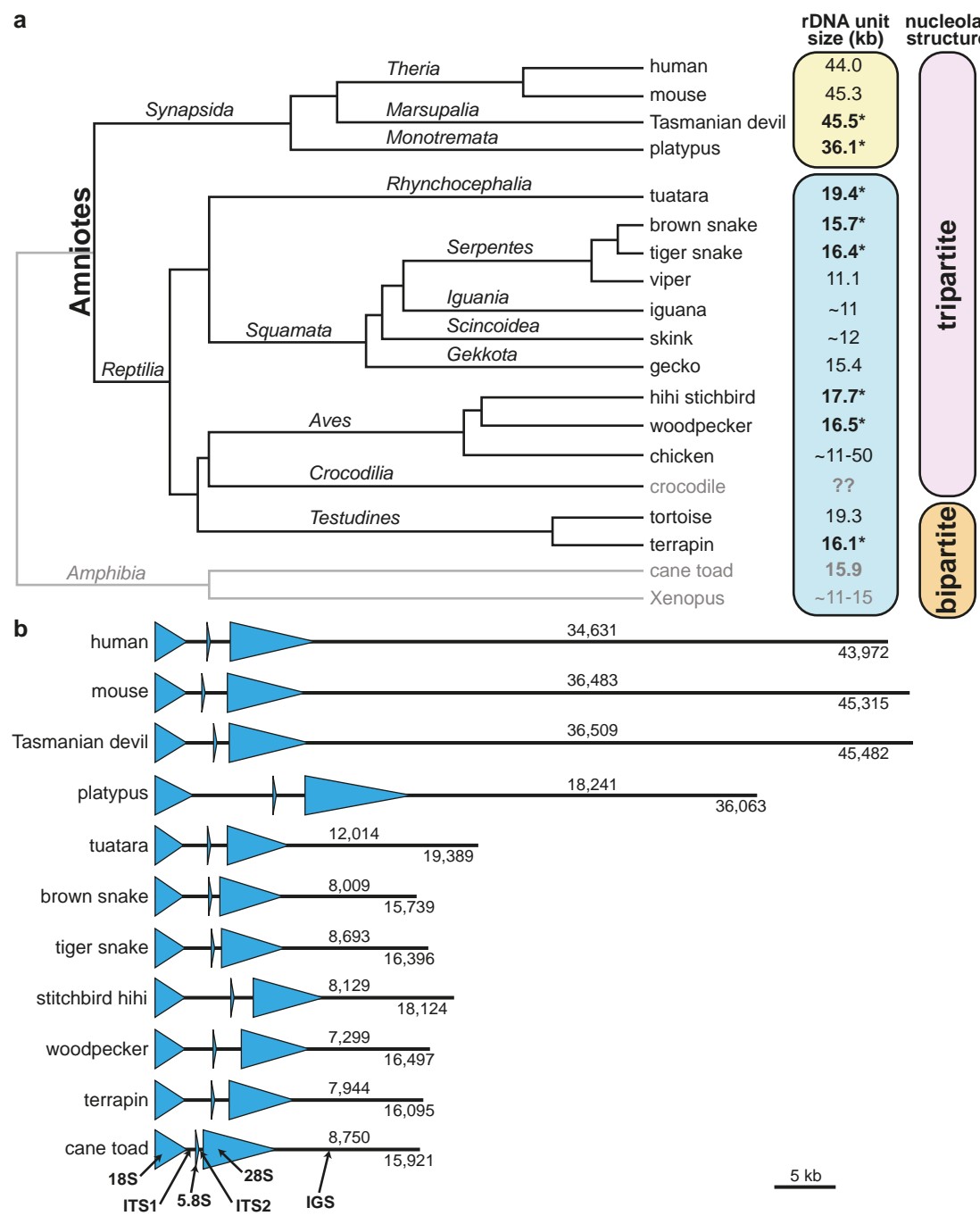

**Fig. 2.** Increase in rDNA size occurred in the ancestor of the mammals. (a) Phylogenetic tree of amniotes showing rDNA unit sizes. Division into clades with "large" (>30 kb; yellow box) vs "normal" (8–20 kb; blue box) rDNA unit sizes is shown to the right, as is division into clades with bipartite vs tripartite nucleolar structure. rDNA unit sizes determined in this study are indicated in bold with asterisks; outgroups are shaded in grey, as is *Crocodilia* for which rDNA unit size data are not available. Some values are indicated as approximate as they were determined with techniques that did not produce precise size measurements. Variation in size is not indicated unless it alters the distinction between normal and large unit size (i.e. chicken). Tree adapted from Gemmell *et al.* (2020). (b) rDNA unit consensus sequence maps, with the 18S, 5.8S, and 28S coding regions (arrowheads) indicated. Numbers above the IGS region are IGS lengths (bp); below and to the right are entire rDNA unit sizes (bp). Diagram is to scale. Size differences from Table 2 are because the values here were calculated from the consensus sequences rather than from the means of the reads. See Table 1 and Supplementary Table 1 for Latin binomials of species and details of species previously determined.

the monotreme, platypus (*Ornithorhynchus anatinus*). PacBio reads are typically not long enough to traverse entire long rDNA units but appear to overcome many limitations in assembling complete rDNA units from short reads, as we have successfully used PacBio reads to assemble entire rDNA units in fungi (Winter *et al.* 2018). To verify that PacBio read assemblies can be used to determine

amniote rDNA unit sizes, we obtained a PacBio assembly for *Malaclemys terrapin* (Diamondback terrapin) from the *Testudines*. Applying the same BLAST method used for ONT data to the contigs from this PacBio assembly, we found 17 contigs that each contained two full rDNA repeats. These rDNA units were ~15.8 kb in size (Supplementary Fig. 4), similar to the size reported for a

**Table 2.** Sizes and variances of amniote rDNA units.

| Species | Number of units | Mean size (bp)[a] | Size standard deviation (bp) |
|---|---|---|---|
| Cane toad | 109 | 15,942 | 1,156 |
| Brown snake | 159 | 15,268 | 799 |
| Tiger snake | 157 | 14,562 | 342 |
| Diamondback terrapin | 17 | 15,740 | 603 |
| Golden-fronted woodpecker | 145 | 16,262 | 580 |
| Stitchbird hihi | 109 | 17,658 | 599 |
| Tuatara | 17[b] | 19,721 | 1,886 |
| Platypus | 9 | 36,015 | 3,143 |
| Tasmanian devil | 147 | 42,127 | 4,377 |

[a]Means and standard deviations calculated from the indicated numbers of ONT units or PacBio assembly contigs (terrapin and platypus). Sizes differ to those reported elsewhere in the manuscript, as those used the consensus units.
[b]Only a small number of reads spanning whole rDNA units were present in the tuatara ONT dataset, hence read number is much smaller than for the other ONT datasets.

*Testudines* species from a different family (*Testudo graeca*; ∼19.3 kb; Cortadas and Pavon 1982). These results show that PacBio assemblies can be used to determine amniote rDNA sizes and confirm that *M. terrapin* has a "normal" rDNA unit size.

We next used the same BLAST method on an existing platypus PacBio assembly (Zhou *et al.* 2021). However, we were unable to find any full-length rDNA units in the assembled contigs, therefore we generated our own PacBio local assembly. We biased the assembly toward the rDNA by using BLAST to identify PacBio reads containing rDNA and then assembling this subset of reads. The assembly resulted in one contig containing a complete rDNA unit. Interestingly, while the total rDNA unit length, at 38.8 kb, is similar to that of other mammals, the platypus IGS is noticeably shorter (22.3 kb vs 34.6 kb in humans for example) but this shorter size is compensated by a much longer ITS1 (6.8 kb vs 1.1 kb in human) (Fig. 2b; Supplementary Table 2). To confirm this unusual platypus rDNA structure, we tried a similar approach with another platypus Pacbio dataset (NCBI accession PRJNA433451). A draft assembly of rDNA-containing reads from this dataset included three contigs that carried complete rDNA units (1, 3, and 4 units). These rDNA units all had the same structure as the previous assembly, but they varied substantially in size, from 30.3 to 40.4 kb (Supplementary Fig. 5). Inspection of the rDNA unit sequences revealed three large sub-repeat arrays in the IGS (repeat unit sizes of ∼585, ∼220 and ∼105 bp) and another in the ITS1 (repeat unit size of ∼610 bp) (Supplementary Fig. 6). Variation in the copy numbers of sub-repeats has previously been shown to cause within-species variation in rDNA unit size (e.g. Rogers and Bendich 1987b; Ganley and Scott 1998; Dyomin *et al.* 2019). Consistent with these other systems, we found that 85–99% of the platypus rDNA unit length variation is a consequence of sub-repeat copy number variation (Supplementary Tables 3 and 4). Thus, platypus has a large rDNA unit size, even though this is achieved in a somewhat different way (relatively small IGS and longer ITS1). Together, these results suggest that large rDNA unit size is a feature of mammals, dating to before the split of the monotremes from the other mammalian lineages.

We calculated the sizes of features within the rDNA unit by annotating rRNA genes onto the reads and onto rDNA consensus sequences that were generated for each of the datasets we analyzed. The results confirmed that the increase in mammalian rDNA unit size results primarily from the expansion of the IGS (Fig. 2b; Supplementary Table 2). Together, our results show that large rDNA unit size cannot be a prerequisite for tripartite nucleolar organization (Fig. 2a) and there is no evidence for it being a consequence of tripartite nucleolar organization.

If a tripartite nucleolar organization cannot explain the large rDNA unit/IGS sizes in mammals, what can? One possibility is a sudden invasion of transposable elements (TEs) into the IGS of a mammalian ancestor, with the resultant large IGS having been passively inherited since this original size expansion. Consistent with this, the IGSs of eutherian mammals characterized to date do contain numerous TEs (Gonzalez and Sylvester 1995; Grozdanov *et al.* 2003; Agrawal and Ganley 2018), while reptilian rDNA units characterized in this study lack TEs, with the exception of a single LTR in the stitchbird hihi IGS. However, we found no TEs in the platypus rDNA and only three in the Tasmanian devil rDNA (two LINEs and one LTR). We wondered if there was some clear alternate route to platypus and Tasmanian devil achieving large rDNA unit sizes other than TE colonization. Given the substantial number of sub-repeats in the platypus rDNA, we looked at the rDNA sub-repeat composition of the other amniote species. We found some species (e.g. brown snake) have no obvious sub-repeats, some (e.g. diamondback terrapin) had degenerate sub-repeats (see Section "Materials and Methods"), while others (e.g. tuatara) have clear sub-repeat arrays (Fig. 3; Supplementary Fig. 7). In contrast to platypus, sub-repeats from the other species were restricted to the IGS. Platypus and Tasmanian devil had the longest total sub-repeat lengths of the species determined here, but the proportion of the IGS occupied by sub-repeats was unremarkable for these species compared to the other species with sub-repeats (Supplementary Table 5). Thus, while sub-repeat length (when present) seems somewhat scaled to total IGS length, sub-repeats alone cannot explain the IGS size of the platypus and Tasmanian devil. We then searched the cognate genomes of platypus and Tasmanian devil for matches >1 kb to the IGS. We found two such matches in the Tasmanian devil, both to unannotated regions that show evidence for transcription (Supplementary Fig. 8). One shows no BLAST or BLASTx match to known elements, while the other has a 34 amino acid match to an unknown protein (Supplementary Fig. 8). These results indicate that duplication of sequences from other parts of the genome into the IGS may have contributed to the Tasmanian devil IGS expansion. Together, however, they suggest that TEs are not a conserved presence in mammalian IGSs, implying they cannot explain the large IGS size of mammals.

It is possible our failure to identify TEs in non-eutherian mammal species is a methodological limitation rather than a true absence. We, therefore, used an alternate approach to look for evidence of an early TE invasion into the mammalian IGS: determine how many TEs in the human and mouse IGSs are orthologous. To do this, we used RepeatMasker to identify TEs in both species' IGSs. However, the number of potentially orthologous TEs was small, with many TEs belonging to different families (Fig. 4). Similar results were obtained regardless of the source species used in RepeatMasker (Supplementary Fig. 9). These results suggest that mammalian IGSs have been independently colonized by TEs and thus mammalian IGS size is not simply a trivial consequence of inheriting and maintaining a large IGS from a mammalian common ancestor. Instead, large IGS size appears to have been maintained over the course of mammalian evolution despite extensive IGS sequence turnover, including TE invasions. Indeed, this sequence turnover is of sufficient magnitude to make the mammalian IGSs examined in this study essentially unalignable (Supplementary Fig. 10). Thus, our results reveal a previously unrecognized selective maintenance of IGS (and/or total rDNA unit) size, a phenomenon that currently lacks explanation.

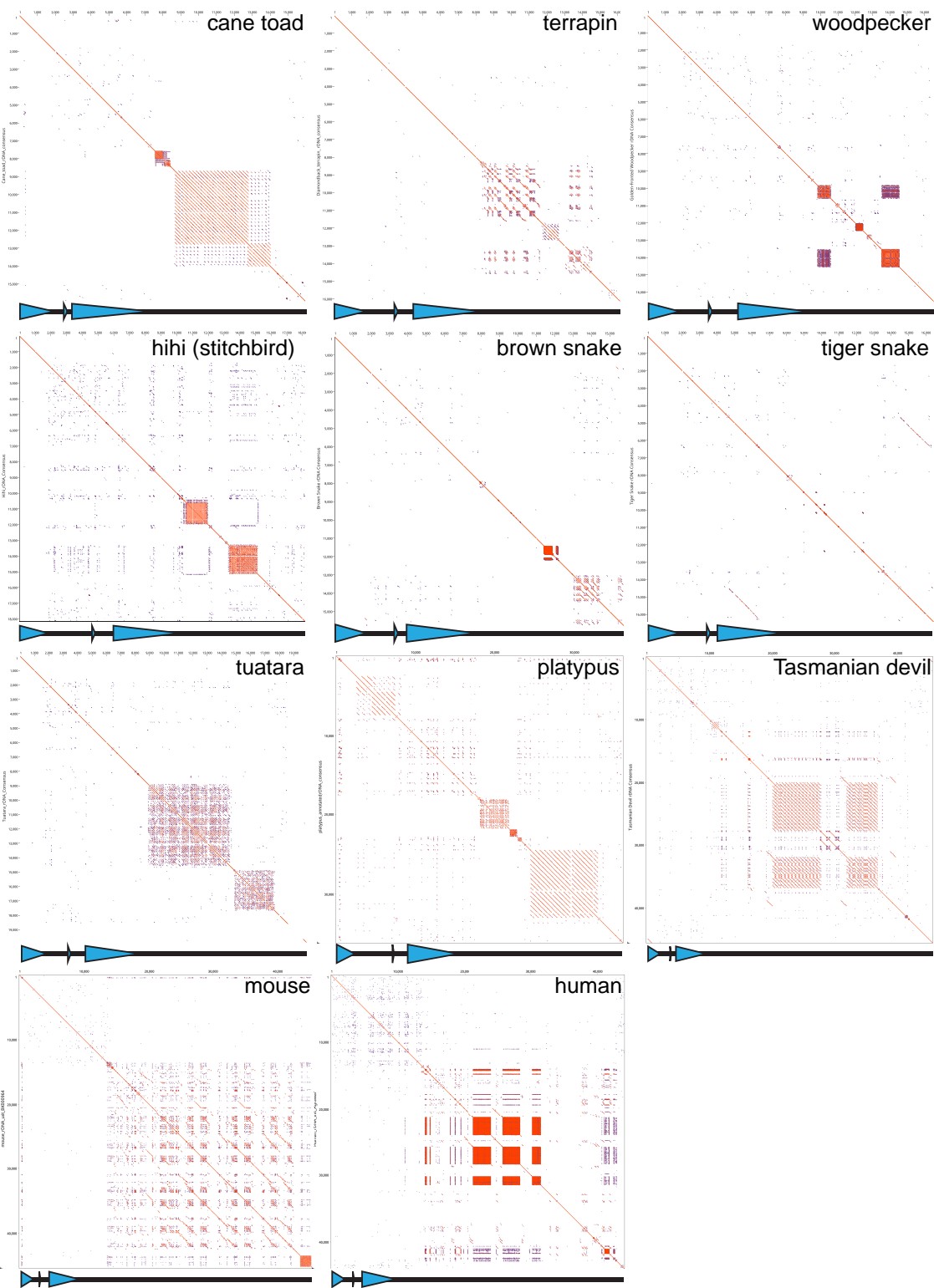

**Fig. 3.** Dotplots of amniote rDNA units showing presence or absence of tandem sub-repeat arrays. Lines indicate sequence matches in an all-vs-all alignment, with sub-repeat arrays appearing as "squares" around the diagonal. Microsatellite arrays also appear as dense squares, for example in human. Schematics of the rDNA units from Fig. 2 are shown below each dotplot, with blue triangles representing rRNA coding regions. Platypus unit is a consensus of the units shown in Supplementary Fig. 6. Mouse rDNA unit is NCBI accession BK000964; human rDNA unit is from Agrawal and Ganley (2018). See Supplementary Fig. 7 for larger versions of the dotplots. Dotplots were created in Geneious (v. 2020.05).

## Discussion

In this study, we used long-read sequence datasets to determine the rDNA unit sizes from representatives of the major amniote clades for which data were available. Our analyses show that all amniote clades we assayed, except mammals, have "normal" rDNA unit sizes (8–20 kb). In contrast, our results suggest that

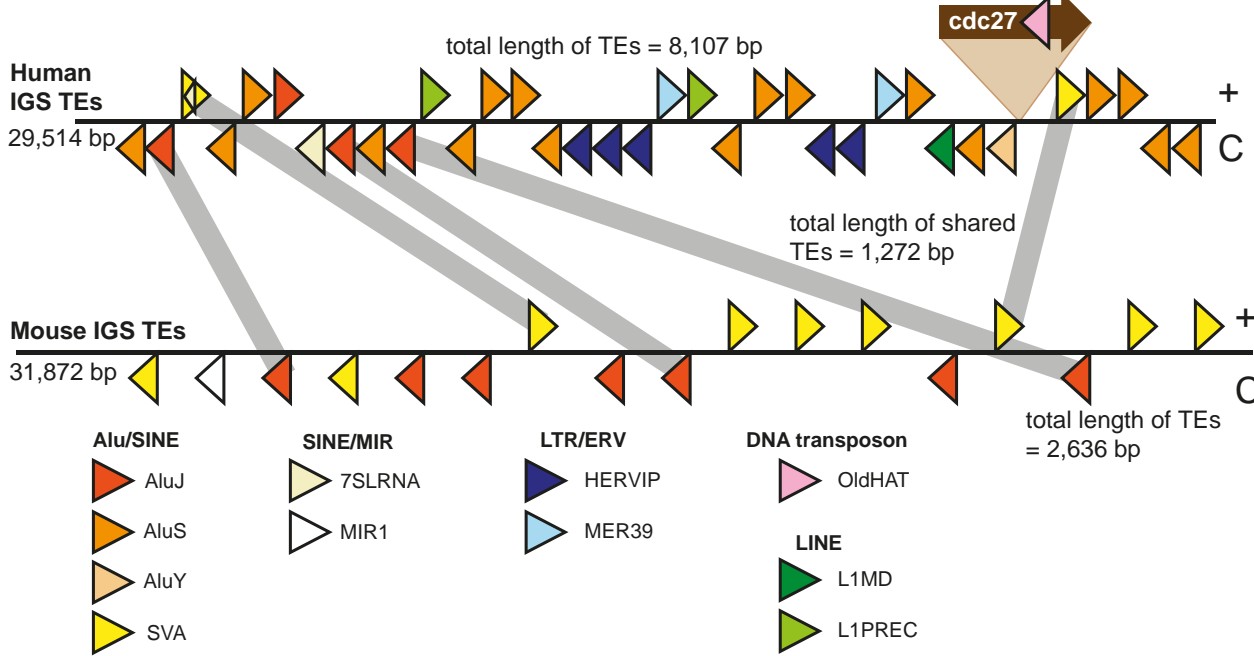

**Fig. 4.** Few IGS TEs are orthologous between human and mouse. TEs detected by RepeatMasker using human as source are indicated schematically by triangles. Colors reflect TE families as indicated below and TE orientation is indicated by triangle direction. Putatively orthologous TEs (determined by conservation of TE type, order and orientation) are indicated by grey bars. Nested TEs are indicated by internal triangles; and a *cdc27* pseudogene found in some primates is illustrated. Total lengths of the IGS, TEs in the IGS, and orthologous TEs are indicated. TE sizing and positioning are not to scale. See Supplementary Fig. 9 for equivalent figure where mouse was used as the RepeatMasker source.

all mammals have large (>30 kb) rDNA units (Supplementary Table 1). Together, these results suggest that large rDNA unit sizes evolved at the base of the mammals, with these large sizes having been maintained since. Our phylogenetic sampling makes the conclusion of large rDNA unit size originating at the base of the mammals robust, even though the number of amniote species with characterized rDNA unit sizes is small. For example, a *Crocodylia* dataset was unavailable, but even if this lineage has long rDNA units the most parsimonious explanation would be an independent increase in rDNA unit size in this lineage, rather than long rDNA units being ancestral. Similarly, our finding of long unit sizes in marsupials and monotremes suggests that even if some mammals are found to have "normal" rDNA unit sizes, this would likely be a secondary reduction in size.

We found that the two bird species examined in this study both have normal rDNA lengths. In comparison, the only previously reported bird rDNA unit length was in chicken, which has a surprisingly large size range that extends up to the large size of mammals (11–50 kb; Delany and Krupkin 1999). The chicken IGS has a high density of sub-repeats (Dyomin *et al.* 2019), higher even than we found here for platypus, and the reported variation in rDNA unit size appears to come from copy number variation of these sub-repeats (Delany and Krupkin 1999; Dyomin *et al.* 2019). Our results suggest that normal rDNA size is ancestral in birds, with chicken extending above this range through sub-repeat expansions. Substantial variation in sub-repeat number causing IGS expansions have occasionally been observed in other groups, for example, in plants (Rogers and Bendich 1987a) and fungi (Ganley and Scott 1998), and as we observed here for platypus. It is unclear if these expansions are a selective response, are neutral by-products of ongoing rDNA recombination perhaps triggered by a loss in control over sub-repeat copy number (Ganley and Scott 1998), or represent selfish evolutionary dynamics (Haig 2021;

Bendich and Rogers 2023). The fact that such expansions are occasionally found across different phylogenetic groups suggests that a more comprehensive determination of rDNA sequences across eukaryotes will unearth further examples, and a broader picture of sub-repeat expansions may provide clues to explain what drives them.

Our results are inconsistent with the proposed link between rDNA unit size and nucleolar structure (Thiry and Lafontaine 2005). Specifically, the tripartite nucleolar structure is observed in all members of the amniotes except the *Testudines* (Thiry *et al.* 2011) and possibly *Rhynchocephalia*. In contrast, our results show a much more limited phylogenetic distribution of large rDNA unit size. This is not surprising, given previous evidence that species with tripartite nucleolar structure do not have long rDNA unit sizes (Cortadas and Pavon 1982; Voronov *et al.* 2008), but our results provide robust phylogenetic support for this conclusion. Thus, increased rDNA unit size cannot have led to tripartite nucleolar structure, nor has been concomitant with its formation. It is possible that tripartite nucleolar organization somehow facilitated the increased mammalian rDNA unit size, but if so it is unclear why rDNA unit size increases have not occurred in other lineages with tripartite nucleolar organization.

The most striking finding from this study is that long rDNA unit size does not appear to have been passively inherited from an expansion event at the dawn of the mammals, but instead has been actively maintained despite widespread IGS sequence turnover. Our conclusion of active maintenance of large mammalian rDNA unit sizes rests on our finding of massive variation in TE presence in mammalian rDNA, including few TEs in platypus and Tasmanian devil, and few orthologous TEs between human and mouse IGSs. These observations do not distinguish between an initial TE invasion into the IGS following by loss of this signal through mutation vs a TE-independent increase in IGS size.

Regardless, the current distribution of TEs in mice and human suggest that numerous transposition and/or deletion events have occurred in the IGSs of these species since their split. Changes in IGS TE composition have been previously reported in primate IGSs (Gonzalez *et al.* 1993; Agrawal and Ganley 2018), and insertion of a *cdc27* pseudogene into the IGS during primate evolution (Gonzalez *et al.* 1993) is further evidence for ongoing IGS sequence turnover. Together, we conclude that mammals have maintained large rDNA unit sizes despite ongoing sequence turnover that has generated marked differences in the sequence compositions of mammalian IGSs, with these conclusions suggesting there is a hitherto unrecognized constraint on IGS length.

We propose two classes of explanations for the active maintenance of increased mammalian rDNA unit size. First, rDNA unit size may result from an upper size limit constraint—a "hard lid" model of rDNA unit size evolution. If that constraint had been relaxed in an early mammal, rDNA unit size could have drifted up to the new limit. However, what form such a constraint could take is unknown. It seems unlikely to simply be total rDNA length as this is also determined by rDNA copy number, which is variable (Hall *et al.* 2022), including between organisms with similar rDNA sizes (e.g. Bik *et al.* 2013). Moreover, no association was found between sub-repeat copy number and rDNA copy number in *Vicia faba* (Rogers and Bendich 1987a). Finally, a hard lid explanation also requires upward pressure so the maximum unit size is observed, and the cause of this putative upward size pressure is similarly unknown.

The other class of explanation is that rDNA unit size is under direct selection for function—an "optimal size" model. What aspect of rDNA could be subject to selection in a way that determines IGS size? One possibility is the number of functional elements in the IGS determines what size it reaches. Consistent with this, a number of IGS elements are conserved across primates and thus may be functional (Agrawal and Ganley 2018). However, this also requires a "hard lid" selection on size, otherwise functional elements only provide a lower bound on IGS size. Consistent with an optimal size explanation, the total length of conserved elements is much larger in the human IGS compared to the yeast IGS. However, the proportion of the IGS that is composed of conserved elements is similar between these two species. Furthermore, conserved elements only comprise a minority of the IGS (compare Agrawal and Ganley (2018) with Ganley *et al.* (2005)) and there is scant evidence for the conservation of IGSs broadly across the mammals. Together, then, there is little evidence that mammals have a sufficiently large set of functional elements to drive consistent, large IGS sizes.

Another possibility is that size is important for some aspects of nucleolar structure. For example, the rDNA appears to have specific conformations in the nucleolus (Maiser *et al.* 2020), and rDNA compaction appears to be important for nucleolar morphology/size (Albert *et al.* 2011). Thus, there may be co-adaptation of rDNA size and nucleolar scaffolding to give optimal nucleolar structure. A related possibility is that the IGS may help modulate nucleolar phase properties, which are involved in nucleolus formation (Lafontaine *et al.* 2021; Mangan and McStay 2021). Finally, the nucleolus is suggested to be a "detention" site where certain proteins are retained until release for use (Audas *et al.* 2012), in which case increased unit size may follow selection for increased detention capacity.

A limitation of these potential explanations is they do not explain why the selected size changed dramatically in a mammalian ancestor. There are few obvious correlates with other changes in the mammalian rDNA—for example, the limited data available suggest rDNA copy number in mammals is unremarkable compared to other amniotes (Long and Dawid 1980; Voronov *et al.* 2008). However, one clue might be that the 5′ external transcribed spacer (5′ETS), which is cleaved from precursor rRNA during rRNA processing (Nazar 2004), is about 5–6 times longer in human and mouse than *Xenopus* (Eichler and Craig 1994). This suggests that 5′ETS size might scale with IGS length. rRNA transcriptional start sites cannot currently be determined from DNA sequence data, so further analysis is required to determine whether a 5′ETS and IGS length correlation does exist.

Finally, the IGS in platypus is curious, as although it has a long rDNA unit, a substantial portion of this consists of sub-repeats. If these sub-repeats are discounted, the rDNA unit size would be in the order of 20 kb. Thus, while we have concluded that platypus has a long rDNA unit size, the results could be interpreted as platypus having a normal rDNA size that appears large because of sub-repeat expansion (similar to chicken; Dyomin *et al.* 2019), or having a transitionary rDNA size between the other mammals and the rest of the amniotes. Determining the rDNA size from a species of the only other extant monotreme family, the *Tachyglossidae* (echidnas), should help determine which of these possibilities is the case, but long-read data are not currently available for any member of this group to our knowledge.

In summary, we have shown that large rDNA unit sizes in mammals appear to result from a size expansion that occurred at the base of the mammals and that this is not directly related to tripartite nucleolar structure as previously suggested. Our results call into question the tacit assumption that the IGS is an approximately neutral region in terms of length and instead suggest there is active maintenance of specific rDNA unit lengths. It remains unclear why mammals have dramatically increased their rDNA unit size. We provide two general models, a "hard lid" model where rDNA unit size results from selection against sizes above a certain threshold, and an "optimal size" model where increased rDNA unit size has been selected for. It is currently difficult to distinguish between these models, but future observational studies and synthetic biology approaches to manipulating the rDNA are likely to be fruitful for improving our understanding of this understudied region of the genome.

## Data availability

The genome projects accessed in this study are shown in Table 1, including accession numbers or contact details as relevant. Supplementary File 1 contains detailed descriptions of all Supplementary files. Supplementary materials not available in Supplementary Figs. or Tables are available through Figshare (10.17608/k6.auckland.25126664), including the consensus sequences and the data used to construct them, and the TEs identified using RepeatMasker. The hihi (stitchbird) data for this study are deposited in the Aotearoa Genomic Data Repository (AGDR) under project ID TAONGA-AGDR00034 (DOI 10.57748/ZD00-D451).

## Supplemental Material

Supplemental material is available at GENETICS online.

## Acknowledgments

We acknowledge the University of Auckland Centre for eResearch and the New Zealand eScience Infrastructure (NeSI; funded jointly by NeSI's collaborator institutions and the Ministry of Business, Innovation & Employment's Research Infrastructure program;

https://www.nesi.org.nz) for high-performance computing facilities. We thank Sylvie Hermann-Le Denmat (University of Auckland) for assistance throughout the project, Anna Santure (University of Auckland) for facilitating access to the hihi data from the AGDR, and Brian McStay (University of Galway) for alerting us to the potential 5′ETS-IGS length correlation. We acknowledge Ngāti Manuhiri as Mana Whenua and Kaitiaki of Te Hauturu-o-Toi and its taonga, including hihi.

## Funding

We acknowledge a Marsden Fund grant (14-MAU-053) to ARDG, Medical Research Futures Fund (MRFF) grants (2016008 and 2016124), and National Health and Medical Research Council (NHMRC) grants (2011277 and 2021172) to HRP, Australian Research Council Discovery Projects (DP210103512 and DP220101429) and NHMRC Ideas grants (2021172 and 2027730) to PDW, and a Genomics Aotearoa High Quality Genomes and Population Genomics Project grant to AW.

## Conflicts of interest

The author(s) declare no conflict of interest.

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

*Editor: S. Edwards*