## [Peer Review File · Genetics]

Origin and maintenance of large ribosomal RNA gene repeat size in mammals

Emma Macdonald, Annabel Whibley, Paul D Waters, Hardip Patel, Richard Edwards, and Austen Ganley

NOTE: The reviews and decision letters are unedited and appear as submitted by the reviewers.

In extremely rare instances and as determined by a Senior Editor or the EIC, portions of a review may be redacted. If a review is signed, the reviewer has agreed to no longer remain anonymous.

The review history appears in chronological order.

Review Timeline:

Submission Date:	2024-03-12
Editorial Decision:	2024-05-02
Resubmission Received:	2024-06-14
Editorial Decision:	2024-07-14
Revision Received:	2024-07-15
Accepted:	2024-07-16

May 2, 2024

GENETICS-2024-306933

Origin and maintenance of large ribosomal RNA gene repeat size in mammals

Dear Dr. Ganley:

Three experts in the field have reviewed your manuscript, and I have read it as well. Overall the reviewers thought your study was novel and noteworthy, and addressed a significant gap in the field. While your manuscript is not currently acceptable for publication in GENETICS, we would welcome a substantially revised manuscript. Both reviewers have comments and concerns to be addressed in a revised manuscript. You can read their reviews at the end of this email.

In general the reviewers wanted more documentation of details of the assemblies as analyzed, as well as some statistical analyses. Reviewer 2 wanted better documentation of transposable elements and rDNA units for some species, to see what rDNA is doing in species with few recognizable TEs. This reviewer also wanted reports on the mean and variance of rDNA number for the species analyzed. Reviewer 1 wanted better justification for the use of ONT sequencing, specifically addressing possible biased errors for that method. This reviewer also wanted some increased clarity on Figure 2 and better representation of bird rDNAs, if possible. It would be great if you could analyze one or two additional birds but I realize this is beyond the scope of the study and is not required for ultimate acceptance. Finally, reviewer 3 wished to see a comparison of your rDNA numbers and those of the T2T genome, as well as more discussion of the evolutionary consequences of rDNA number. This last point is a bit vague and so need not be addressed, although it could be a good opportunity to make your paper more impactful.

We look forward to receiving your revised manuscript. Please let the editorial office know approximately how long you expect to need for revisions.

Upon resubmission, please include:

1. A clean version of your manuscript;
2. A marked version of your manuscript in which you highlight significant revisions carried out in response to the major points raised by the editor/reviewers (track changes is acceptable if preferred);
3. A detailed response to the editor's/reviewers' feedback and to the concerns listed above. Please reference line numbers in this response to aid the editor and reviewers.

Your paper will likely be sent back out for review.

Additionally, please ensure that your resubmission is formatted for GENETICS

<https://academic.oup.com/genetics/pages/general-instructions>

Follow this link to submit the revised manuscript: Link Not Available

Sincerely,

Scott Edwards
Associate Editor
GENETICS

Approved by:
David Begun
Senior Editor
GENETICS

Reviewer #1 (Comments for the Authors (Required)):

Summary

This manuscript interrogated two primary research questions about the phylogenetic evolution of rDNA size: 1) what is the origin of large rDNA unit size in mammals, and 2) how are the larger rDNA unit sizes maintained since the origin. Across a phylogenetic tree of representative Amniotes, the authors compared rDNA unit sizes obtained from analyses of published and unpublished long-read sequencing data, as well as published records of rDNA unit sizes. They confirmed increases in intergenic

spacer (IGS) length as the major contributor of mammalian size increase. From the contrast between larger rDNA unit sizes in investigated Synapsida species and smaller sizes in Reptilia, they concluded that rDNA unit size increased at the base of the mammalian clade, and the increase is not associated with tripartite nucleolar organization. Given the high sequence turnover of rDNA sequences (i.e., lack of TE sequence sharing) across the investigated mammalian species, the authors proposed that an explanatory model, that a large increase first in IGS occurred in a mammalian ancestor, and was then maintained independently in different mammals (e.g., via selection).

The present study did a great job capturing the knowledge gap in rDNA size evolution, throwing out intriguing research questions, and providing clear answers. The paper is also well-written and highly readable. Although the amount of work may not be as substantial as studies that normally publishes at Genetics, its novelty and significance satisfies the scope of this journal.

Nevertheless, I have a few major concerns regarding whether the presented data can support conclusions, as well as rigor of data presentation and interpretation. These concerns need to be addressed to meet the standards of publication.

Major concerns & suggestions

Comparability of rDNA unit size across different types of assays

It is controversial that the authors emphasized the benefits of ONT on improving the estimation of rDNA unit size, but without discussing to what extent the rDNA sizes calculated from the improved data (ONT) are comparable with those from other assays (PacBio, NGS, southern blots). It is unclear to me if there are known or unknown variance added to the estimation of the actual size distribution, and/or if there are directional biases in the estimation from some of those assays (the worse case). While the authors have already labeled source of data in Figure 2A, they should either clarify the above questions, which could be challenging without existing data, or at least acknowledge the risks of cross-platform data comparisons (even if no statistical tests were done) in Discussions.

Presentation of rDNA unit size distribution

The presentation of variation in within-species rDNA unit sizes is highly inconsistent in Figure 2A. For different species, the authors used single values, approximate values, and ranges (e.g., chicken). The minimum standard should include mean value and standard deviation. While historical records of rDNA unit size for some species may lack distribution information, it is more appropriate to either only present consistent data in the main figures, or to re-calculate rDNA distribution from recent sequencing data.

Is Chicken an exception of 'normal' rDNA sizes categories?

The data cannot support if "chicken extends above the range of other birds simply through sub-repeat expansions", or if there's interesting evolution, unless the authors can calculate the rDNA sizes of chicken (and other birds presented) with sub-repeats discounted, similar to platypus. So I think it is better to list both possibilities. Even though the authors already listed one of them and used 'likely', it sounds like they are suggesting an unsubstantiated conclusion. In biology, it is fine to maintain exceptions that cannot be well-explained.

Minor comments

Line 270: Current data cannot prove single evolutionary event: While data presented in the study can narrow down the origin of rDNA unit size increase to the base of the mammals, we cannot rule out the possibility of multiple events of such increases at the base of the mammals is unknown.

Figure 2A: the bold text is not clear when shaded in grey (i.e., for "cane toad"). An alternative is to star long-read sequenced species with a "*".

Figure 2A: "Synaspida" should be corrected to "Synapsida"

Reviewer #2 (Comments for the Authors (Required)):

This study uses long-read sequencing to investigate rDNA unit length among amniotes. The authors find a dramatic size increase that is shared across mammals, but not present in other amniotes, except for what seems to be high variation within chicken. The phylogenetic distribution of size increase is not consistent with the leading hypothesis for rDNA size increase (tripartite nucleolar structure). Further, the intergenic spacer region that comprises the bulk of the length increase does not appear to be conserved at the sequence level, suggesting that size itself that may be the target of selection. Multiple potential

hypotheses are proposed but will require additional studies to address.

I found this manuscript to be very well-presented and thought-provoking. The data seem clear and although lineages remain sparsely sampled, the conclusions seem straightforward based on their analyses. I think this will be of interest to many readers of Genetics.

My only concern is a general lack of statistical analysis/presentation. For instance, I think the manuscript would be strengthened if results from the database searches for ONT reads with multiple rDNA units (L119-122) were more comprehensively reported. The supplemental figure is useful, but for instance, it would be nice to have some idea for the mean, variance, and number of rDNA units measured in each species?

Finally, I was left wondering, "What is the sequence content of species with large rDNA size but no (or few) recognizable TEs (e.g. Tasmanian devil and platypus)?" Perhaps the authors could expand on their analysis/discussion of how large rDNA unit size could have been maintained without TE involvement.

Reviewer #3 (Comments for the Authors (Required)):

The manuscript measured rDNA unit size in the major amniote lineages using publicly available and unpublished long-read sequence data to determine when large IGS sizes evolved and whether they are associated with tripartite nucleolar organization. This study is of great significance for studying the sequence structure of rDNA in different species and the role of rDNA in evolution.

Some of my comments are as follows:

1. Have the authors compared the differences between the human rDNA sequence assembled by the author and the published T2T genome sequence?
2. Please add a discussion about the possible impact of rDNA copy number variation on evolution in the manuscript.

Associate Editor Comments:

Responses to Reviewers

Reviewer #1 (Comments for the Authors (Required)):

Major concerns & suggestions

Comparability of rDNA unit size across different types of assays:

It is controversial that the authors emphasized the benefits of ONT on improving the estimation of rDNA unit size, but without discussing to what extent the rDNA sizes calculated from the improved data (ONT) are comparable with those from other assays (PacBio, NGS, southern blots). It is unclear to me if there are known or unknown variance added to the estimation of the actual size distribution, and/or if there are directional biases in the estimation from some of those assays (the worse case). While the authors have already labeled source of data in Figure 2A, they should either clarify the above questions, which could be challenging without existing data, or at least acknowledge the risks of cross-platform data comparisons (even if no statistical tests were done) in Discussions.

We do not claim that ONT is the best technology for determining rDNA unit length, but simply that long-read approaches are superior to short-read approaches as it is usually not possible to construct an entire rDNA unit from short-read data alone. Thus, we modified the sentence starting line 117 to read "*Long-read sequencing approaches can overcome these limitations of short-read sequencing because they generate reads that can traverse most or all the rDNA unit length.*" to try and emphasize that long-read approaches are better than short-read, not necessarily the best overall method.

We already went to some lengths to show that both the ONT and PacBio approaches we employed could produce comparable rDNA unit sizes to those determined by other techniques. While our results cannot rule out some bias, if it does exist our results suggest it is not of sufficient magnitude to impact whether an rDNA unit is seen as normal or large. The two sections in the text presenting this evidence are copied here for convenience:

Line 123 - "*To verify that this approach could accurately determine rDNA unit sizes, we tested it with species known to have 'normal' (~9 kb; Saccharomyces cerevisiae) and large (~44 kb, Homo sapiens) rDNA units. As expected, rDNA unit size was accurately determined for both species (Figure S1).*"

Line 200 - "*To verify that PacBio read assemblies can be used to determine amniote rDNA unit sizes, we obtained a PacBio assembly for Malaclemys terrapin (Diamondback terrapin) from the Testudines. Applying the same BLAST method used for ONT data to the contigs from this PacBio assembly, we found 17 contigs that each contained two full rDNA repeats. These rDNA units were ~15.8 kb in size*

(Figure S4), similar to the size reported for a Testudines species from a different family (Testudo graeca; ~19.3 kb; CORTADAS AND PAVON 1982)."

Presentation of rDNA unit size distribution

The presentation of variation in within-species rDNA unit sizes is highly inconsistent in Figure 2A. For different species, the authors used single values, approximate values, and ranges (e.g., chicken). The minimum standard should include mean value and standard deviation. While historical records of rDNA unit size for some species may lack distribution information, it is more appropriate to either only present consistent data in the main figures, or to re-calculate rDNA distribution from recent sequencing data.

We certainly understand the reviewer's point here. As the reviewer notes, the uncertainty we indicated in most cases is because the size estimates from some old methods were approximate. On balance, we would prefer to keep the information in this figure despite the inconsistency for two reasons. First, it better places our contribution in the context of what is already known, including the overall consistency of our results with the older studies. Second, we think that including such old, less precise data emphasizes the extent to which amniote rDNA unit size has been ignored. We considered the reviewer's suggestion of presenting distributions for all taxa, but decided against this for two reasons. First, size variation could come from intra-genomic, inter-cellular and/or inter-individual variation, and it would be hard to make the distributions for different species comparable across all these potential sources of variation. Thus, we think that some inconsistency in presentation is less potentially damaging than making assumptions about the comparability of the ranges presented that are difficult to validate. Second, for the main goal of the paper, small fluctuations in unit size (whether real or artefacts) do not affect our conclusions. For this latter reason, we also removed the small variation in gecko unit size from Figure 2. Finally, we modified the Figure 2 legend to include the following wording, aimed to avoid misleading readers regarding these distributions:

"Some values are indicated as approximate as they were determined with techniques that did not produce precise size measurements. Variation in size is not indicated unless it alters the distinction between normal and large unit size (i.e. chicken)."

This comment also relates to the first comment from Reviewer 2 regarding statistical presentation of the data. In response to that comment we added better statistical description of the ranges of rDNA unit size we estimate, including a new Table 2, again emphasizing that the variation we see does not impact our ability to distinguish normal versus large rDNA units. See response to this comment below for further details.

Is Chicken an exception of 'normal' rDNA sizes categories?

The data cannot support if "chicken extends above the range of other birds simply through sub-repeat expansions", or if there's interesting evolution, unless the authors can calculate the rDNA sizes of chicken (and other birds presented) with sub-repeats discounted, similar to platypus. So I think it is better to list both possibilities. Even though the authors already listed one of them and used 'likely', it sounds like they are suggesting an unsubstantiated conclusion. In biology, it is fine to maintain exceptions that cannot be well-explained.

This is an excellent point, and we think the problem arose partly because we did not sufficiently separate out two different ideas - the origin of long units versus what is happening in birds. To rectify this, we pulled the discussion on birds from the first, origin-based paragraph of the Discussion and incorporated it into a new, second paragraph that discusses what is happening in birds (starting line 384), including comparing to other documented examples. In doing so, we have not advocated for one view, but instead point out the possible explanations for sub-repeat expansions and one potential route to resolving this:

"We found that the two bird species examined in this study both have normal rDNA lengths. In comparison, the only previously reported bird rDNA unit length was in chicken, which has a surprisingly large size range that extends up to the large size of mammals (11-50 kb; DELANY AND KRUPKIN 1999). The chicken IGS has a high density of sub-repeats (DYOMIN et al. 2019), higher even than we found here for platypus, and the reported variation in rDNA unit size appears to come from copy number variation of these sub-repeats (DELANY AND KRUPKIN 1999; DYOMIN et al. 2019). Our results suggest that normal rDNA size is ancestral in birds, with chicken extending above this range through sub-repeat expansions. Substantial variation in sub-repeat number causing IGS expansions have occasionally been observed in other groups, for example in plants (ROGERS AND BENDICH 1987a) and fungi (GANLEY AND SCOTT 1998), and as we observed here for platypus. It is unclear if these expansions are a selective response, are neutral by-products of ongoing rDNA recombination perhaps triggered by a loss in control over sub-repeat copy number (GANLEY AND SCOTT 1998), or represent selfish evolutionary dynamics (HAIG 2021; BENDICH AND ROGERS 2023). The fact that such expansions are occasionally found across different phylogenetic groups suggests that more comprehensive determination of rDNA sequences across eukaryotes will unearth further examples, and a broader picture of sub-repeat expansions may provide clues to explain what drives them."

Minor comments

Line 270: Current data cannot prove single evolutionary event: While data presented in the study can narrow down the origin of rDNA unit size increase to the base of the mammals, we cannot rule out the possibility of multiple events of such increases at the base of the mammals is unknown.

Yes, we agree. Therefore, we reworded the two relevant sentences (starting line 374) to now read:

“Together, these results suggest that large rDNA unit size evolved at the base of the mammals, with these large sizes having been maintained since. Our phylogenetic sampling makes the conclusion of large rDNA unit size originating at the base of the mammals robust...”

Figure 2A: the bold text is not clear when shaded in grey (i.e., for "cane toad"). An alternative is to star long-read sequenced species with a "*".

We have added asterisks as suggested (and updated the figure legend to reflect this).

Figure 2A: "Synaspida" should be corrected to "Synapsida"

Thanks very much for catching this - corrected now.

Reviewer #2 (Comments for the Authors (Required)):

My only concern is a general lack of statistical analysis/presentation. For instance, I think the manuscript would be strengthened if results from the database searches for ONT reads with multiple rDNA units (L119-122) were more comprehensively reported. The supplemental figure is useful, but for instance, it would be nice to have some idea for the mean, variance, and number of rDNA units measured in each species?

This is a good point. We have now made a new table, which we believe is worth making a table in the main text, so this is now Table 2. It includes all the elements suggested by the Reviewer. In addition, we have now described the variation in rDNA unit size between units by adding the following text starting on line 162:

*“We observed variation in rDNA unit sizes between reads, which may result from technical variation due to errors during ONT sequencing (MIKHEENKO et al. 2022) and/or real intra-genomic repeat size variation (e.g. COEN et al. 1982; DE LUCCHINI et al. 1988). Nevertheless, the standard deviation was less than 10% of unit size for all species except Tasmanian devil (**Table 2**), and this variation does not confound the categorization into normal and large size classes.”*

Finally, I was left wondering, "What is the sequence content of species with large rDNA size but no (or few) recognizable TEs (e.g. Tasmanian devil and platypus)?" Perhaps the authors could expand on their analysis/discussion of how large rDNA unit size could have been maintained without TE involvement.

The short answer is we don't know. However, we looked into it, initially by looking in more detail at the sub-repeats. These might explain some of the expansion, but certainly not all of it, and we created a new supplementary table (S5) to show the relative contributions of sub-repeats to IGS size across the amniotes analyzed here. This also provided a more natural home for presenting the sub-repeat results, leading us to make a new figure (3) and a new supplementary figure (S7) from what was originally in the Appendix to reflect this greater relevance.

We then checked whether the Tasmanian devil and/or platypus IGS have elements duplicated from other parts of their respective genomes. We found two such regions in Tasmanian devil, but it is not clear there is any insightful link from these, other than an indication of duplications into the IGS. We made a new supplementary figure (S8) to document this. As a result, the paragraph starting on line 299 has now been substantially expanded (and a description of the additional methods used has been added to that section), with the new text in the Results reading as follows:

*“We wondered if there was some clear alternate route to platypus and Tasmanian devil achieving large rDNA unit sizes other than TE colonization. Given the substantial numbers of sub-repeats in the platypus rDNA, we looked at the rDNA sub-repeat composition of the other amniote species. We found some species (e.g. brown snake) have no obvious sub-repeats, some (e.g. diamondback terrapin) had degenerate sub-repeats (see **Materials and Methods**), while others (e.g. tuatara) have clear sub-repeat arrays (**Figure 3; Figure S7**). In contrast to platypus, sub-repeats from the other species were restricted to the IGS. Platypus and Tasmanian devil had the longest total sub-repeat lengths of the species determined here, but the proportion of the IGS occupied by sub-repeats was unremarkable for these species compared to the other species with sub-repeats (**Table S5**). Thus, while sub-repeat length (when present) seems somewhat scaled to total IGS length, sub-repeats alone cannot explain the IGS size of platypus and Tasmanian devil. We then searched the cognate genomes of platypus and Tasmanian devil for matches >1 kb to the IGS. We found two such matches in the Tasmanian devil, both to unannotated regions that show evidence for transcription (**Figure S8**). One shows no BLAST or BLASTx match to known elements, while the other has a 34 amino acid match to an unknown protein (**Figure S8**). These results indicate that duplication of sequences from other parts of the genome into the IGS may have contributed to the Tasmanian devil IGS expansion.”*

Reviewer #3 (Comments for the Authors (Required)):

The manuscript measured rDNA unit size in the major amniote lineages using publicly available and unpublished long-read sequence data to determine when large IGS sizes evolved and whether they are associated with tripartite nucleolar organization. This study is of great significance for studying the sequence structure of rDNA in different species and the role of rDNA in evolution.

Some of my comments are as follows:

1. Have the authors compared the differences between the human rDNA sequence assembled by the author and the published T2T genome sequence?

We hadn't, but have done so now and have added the result to Figure S1 and added the following text to the manuscript starting at line 126:

“We also compared the rDNA unit sequence we obtained to that from the recent telomere-to-telomere CHM13 human genome assembly, as this included ONT reads (NURK et al. 2022). The rDNA unit sequences were 85% identical, with the most substantive differences being in Long Repeat regions, and the units are structurally the same (Figure S1). The differences may be a consequence of real sequence variation that the CHM13 assembly was able to resolve by separately assembling rDNA from different chromosomes using a local assembly approach and/or may reflect limitations of our low-coverage approach. Together, these results suggest that our ONT read strategy is capable of accurately determining rDNA unit sizes, but that dedicated, high-coverage approaches such as those employed by the T2T consortium (NURK et al. 2022) are likely needed for accurate assembly of rDNA sequences..”

2. Please add a discussion about the possible impact of rDNA copy number variation on evolution in the manuscript.

It was not clear to us in what context we should be discussing rDNA copy number variation - in what way it might be relevant to the rDNA unit size story of this manuscript. We considered a couple of different ways it might be relevant, and thus added the following to the text to point this out, one starting on line 506:

“Moreover, no association was found between sub-repeat copy number and rDNA copy number in Vicia faba (ROGERS AND BENDICH 1987a)”

And the other starting on line 538:

“There are few obvious correlates with other changes in the mammalian rDNA - for example, the limited data available suggest rDNA copy number in mammals is unremarkable compared to other amniotes (LONG AND DAWID 1980; VORONOV et al. 2008).”

July 14, 2024

RE: GENETICS-2024-307168

Dear Dr. Ganley:

I am pleased to accept your manuscript entitled "Origin and maintenance of large ribosomal RNA gene repeat size in mammals" for publication in GENETICS, pending minor revision.

Please submit your revision along with a brief description of how you modified the manuscript in response to the reviewers' concerns and suggestions (which can be viewed at the bottom of this email. Most important are revising the figure 3 as suggested by reviewer 2. I expect you should be able to submit a revised manuscript within 30 days. A suitably revised manuscript will be acceptable for publication; I don't expect to send it out for review.

Please ensure that you have included a Data Availability Statement at the end of the Materials and Methods section. Details available at <https://academic.oup.com/genetics/content/prep-manuscript>. The DAS should include the accession numbers or DOIs of any data you have placed in public repositories, describe supplemental material, include applicable IRB numbers, and may include specifications for how to properly acknowledge or cite the data.

Please follow this link to submit the revised manuscript: Link Not Available

Thank you for submitting this story to Genetics.

Sincerely,

Scott Edwards
Associate Editor
GENETICS

Approved by:
David Begun
Senior Editor
GENETICS

Reviewer comments:

Reviewer #1 (Comments for the Authors (Required)):

In the revised manuscript, the authors have done a great job addressing most of my concerns. This manuscript will be almost ready for publication, as long as they can better address one of my primary concern about the comparability of rDNA unit size across different types of assays:

I guess my point here was not clear to the authors: it is most important that the rDNA size estimated from long-read sequencing is comparable to those estimated from other techniques, if they are still presenting those historical data.

I would expect that data from ONT can make accurate estimates, and results from ONT is comparable to that PacBio. But the authors must either clarify how comparable are the rDNA sizes calculated from ONT/PacBio to those from previously published sizes that were estimated using NGS or older methods, or admit the limitation if they are not sure (which is fine).

Reviewer #2 (Comments for the Authors (Required)):

The authors have addressed my few comments/concerns from the initial review. I noticed a minor typo in the new Figure 3 ("stitchbird" should be "stitchbird"). Also, optionally in all the figures that include dot plots, a more precise context for the statement "coding region is to the left; the IGS to the right" could be provided by illustrating the gene structure (as in Figure 2)

near the axes, even as an approximate location of coding and noncoding regions.

Reviewer #3 (Comments for the Authors (Required)):

all my concerns have been addressed.

Associate Editor comments:

Responses to Reviewers

Reviewer #1

We added the following sentence to the third paragraph of the results:

“The sizes we estimate are comparable to those from made previously from other reptilian species, suggesting these older size estimates made using restriction-hybridization methods (**Table S1**) are comparable to those made from sequence data, as previously indicated (e.g. Gonzalez and Sylvester 1995).”

Reviewer #2

We corrected Figure 3 stitchbird spelling and added schematics of rDNA units from Figure 2 below each dotplot as requested. We also added these schematics below each dotplot in Figure S7. Both legends were revised accordingly.

Data availability

We have included a data availability statement that details the Figshare location of data not available in the manuscript or supplementary material. We trust this is appropriate, but are happy to update if not.

July 16, 2024

RE: GENETICS-2024-307168R1

Dr. Austen R.D. Ganley
University of Auckland
School of Biological Sciences
3A Symonds St, Building 110N
Auckland, N/A 1142
New Zealand

Dear Dr. Ganley:

Congratulations! We are delighted to inform you that your manuscript entitled "Origin and maintenance of large ribosomal RNA gene repeat size in mammals" is acceptable for publication in GENETICS. Many thanks for submitting your research to the journal and for revising your manuscript in light of the reviewer comments.

To Proceed to Production:

1. Format your article according to GENETICS style, as discussed at <https://academic.oup.com/genetics/pages/general-instructions>, and upload your final files at <https://genetics.msubmit.net>.
2. Your manuscript will be published as-is (unedited-as submitted, reviewed, and accepted) at the GENETICS website as an Advanced Access article and deposited into PubMed shortly after receipt of source files and the completed license to publish. Please notify sourcefiles@thegsajournals.org if you do not wish to publish your article via Advanced Access.
3. We invite you to submit an original color figure related to your paper for consideration as cover art. Please email your submission to the editorial office or upload it with your final files. You can submit a small-sized image for evaluation, and if selected, the final image must be a TIFF file 2513px wide by 3263px high (8.375 by 10.875 inches; resolution of 600ppi). Please avoid graphs and small type.

If you have any questions or encounter any problems while uploading your accepted manuscript files, please email the editorial office at sourcefiles@thegsajournals.org.

Sincerely,

Scott Edwards
Associate Editor
GENETICS

Approved by:
David Begun
Senior Editor
GENETICS

note: Please add jnls.author.support@oup.com and genetics.oup@kwglobal.com (or the domains @oup.com and @kwglobal.com) to your email program's "safe senders" list. You will be contacted by both at various points during the production process.

Review comments (if applicable):